# ARE YOU CERTAIN THAT IT IS A DEEPFAKE? DETECTION, GENERATION, AND SOURCE DETECTION FROM AN UNCERTAINTY PERSPECTIVE

## ABSTRACT

As generative models are advancing in quality and quantity for creating synthetic content, deepfakes begin to cause online mistrust. Deepfake detectors are proposed to counter this effect, however, misuse of detectors claiming fake content as real or vice versa further fuels this misinformation problem. In this paper, we evaluate, compare, and analyze the *uncertainty* of these deepfake detectors. As reflected in detectors' responses, deepfake generators also contribute to this uncertainty as their generative residues vary, so we cross the uncertainty analysis of deepfake detectors and generators. We evaluate uncertainty on two datasets with nine generators, with four blind and two biological detectors, compare different uncertainty methods, and perform ablation studies. In addition to image- and region-based uncertainty analysis, we propose novel *uncertainty maps* to decipher the relation between generative artifacts and detector response, also contrasting to detector explainability. We conduct binary real/fake, multi-class real/fake, source detection, and leave-one-out experiments between generator/detector combinations to document their uncertainty, generalization capability, model calibration, and robustness against adversarial attacks. This comprehensive, uncertainty-forward analysis addresses a critical gap in current deepfake detection understanding and thus restore trust in media in the age of generative AI.

## 1 INTRODUCTION

Recently, synthetic content has become a part of our daily lives with the proliferation of generative models. Specifically, human faces have always been the focus of computer vision algorithms, pursuing the same paradigm with generative models since Generative Adversarial Networks (Goodfellow et al., 2014) (GAN) in 2014. In the intersection was born deepfakes: images, audio clips, or videos, where the actor or the action of the actor is fabricated using deep generative models.

Although synthetic content creation brought up many positive use cases, deepfakes are usually exploited in politics, entertainment, and security (Dee, a;b); causing the need of a line of defense (Chu et al., 2020). Deepfake detectors are proposed to satisfy this need, however, their generalization and robustness vary depending on signals, models, and datasets they utilize. As opposed to traditional evaluations based on accuracy and AUC metrics, we analyze their core performance by uncertainty analysis, which has not garnered much attention in the research community. As detectors are being deployed into real-world decision mechanisms, understanding, quantifying, and mitigating the uncertainty associated with their predictions become essential. Understanding the ecosystem of deepfake detectors and generators help assess how they overfit or generalize, model authenticity or artifacts, perform on unseen distributions, handle adversarial attacks, and support source detection.

Deepfake detectors use the existence or non-existence of priors to classify content as real or fake, where these priors are either irreproducible authentic signals (e.g., corneal reflections (Hu et al., 2021), blood flow (Çiftçi et al., 2020a), phoeneme-viseme mismatches (Agarwal et al., 2020)) in real data or small generative artifacts in fake data (Afchar et al., 2018; Chollet, 2017; Coccomini et al., 2022). As per generative models, face generation (Karras et al., 2019; Rombach et al., 2022), face swapping (Fac, b; Li et al., 2019), and face reenactment (Prajwal et al., 2020; Thies et al., 2016) methods create faces using different techniques operating on different facial regions. As a result, dif-

ferent generative models leave different residual traces behind (Wang et al., 2020). Those traces are directly correlated to the detector response, tying deepfake detectors and generators. This connection resumes and is observable in the uncertainty estimation of detectors, aggregated by model-specific uncertainty contributors such as architecture, signal manifold, and training data.

In this paper, we analyze uncertainty of various deepfake detectors on data generated by various deepfake generators. We use this analysis comprehensively to compare the robustness and reliability of detectors, to explain the detector response towards different generative sources, and to perform source detection of deepfakes. As the uncertainty can stem from multiple sources, we also quantify uncertainty using different approaches. Our contributions include,

- in-depth uncertainty analysis of deepfake detectors with respect to deepfake generators,
- understanding uncertainty of source detection to classify generative model of a deepfake,
- image-, region-, and pixel-wise uncertainty comparisons of authenticity- and fakery-based deepfake detectors, and
- explainability-backed uncertainty estimation of generative artifacts.

We conduct our experiments on two datasets, nine generators, six detectors (including four blind and two biological detectors), and two uncertainty methods. We relate generator properties to detector predictions through predictive and model uncertainty using Bayesian Neural Networks (BNNs) and through model uncertainty based on model variance using Monte-Carlo (MC) dropout. We compare and contrast uncertainties on traditional deepfake detection, the more elaborate deepfake source detection, and leave-one-out detection tasks. We perform ablation studies on uncertainty estimation parameters. We measure detector uncertainty on adversarial images to explore robustness. Finally, we formulate novel uncertainty maps to intersect our uncertainty analysis with explainability methods, visualizing the big picture of detector-generator ecosystem.

## 2 RELATED WORK

### 2.1 DEEPFAKE GENERATION

Deepfakes have been increasing in quality and quantity (Mirsky & Lee, 2021), mainly created (1) from scratch with learned distributions (Choi et al., 2018; Karras et al., 2019; Demir & Çiftçi, 2021a), (2) by partial or full face transfer (Fac, a; Dee, c; Li et al., 2019), or (3) by expression or lip reenactment (Prajwal et al., 2020; Thies et al., 2015; 2019). Historically, autoregressive models (Van den Oord et al., 2016) (AR), Variational Autoencoders (Bao et al., 2017) (VAE), Generative Adversarial Networks (Goodfellow et al., 2014) (GAN), or diffusion models (Rombach et al., 2022) are used to create such manipulated content; all of which leave behind different generative residues based on the architecture, the noise, and the operations (Wang et al., 2020).

### 2.2 DEEPFAKE DETECTION

The arms race between generation and detection intensifies as it becomes impossible to distinguish deepfakes from real faces (Tolosana et al., 2020). Deepfake detectors first focused on *artifacts of fakery*, learning directly from data with "blind" detectors which do not exploit any intermediate signal or transformation (Afchar et al., 2018; Chollet, 2017; Zhou et al., 2017; Li et al., 2020; Tariq et al., 2018; Zhou et al., 2017; Khodabakhsh et al., 2018; Güera & Delp, 2018; Nguyen et al., 2019; Barni et al., 2017; Guarnera et al., 2020; Amerini et al., 2019). Although they provide high accuracy on small datasets; they tend to overfit, they are easily manipulated by adversarial samples, and their generalization is limited across different domains, image transformations, and compression levels(Saremsky et al., 2022; Carlini & Wagner, 2017).

Another branch of deepfake detection explores authenticity signals, mostly hidden in biometric data. These detectors explore low to high level signals such as blinks (Li et al., 2018), blood flow (Çiftçi et al., 2020a), head-pose (Yang et al., 2019), emotions (Hosler et al., 2021), gaze (Demir & Çiftçi, 2021b), and breathing (Korshunov & Marcel, 2018). These signals tend to be much inconsistent in fake videos, so the preservation of spatial, temporal, and spectral features in real videos provide an advantage to these detectors for generalization over blind detectors. However, some of these inconsistencies are easily "fixed" in newer generative models (Ruzzi et al., 2023).

The third and newest branch of deepfake detection aims to trace back the source generative model behind a given synthetic sample (Yu et al., 2019; Marra et al., 2019; Çiftçi et al., 2020b; Ding et al., 2021; Çiftçi & Demir, 2022), following the hidden generative residue of the deep models. Some approaches even try to infer model parameters from these artifacts (Asnani et al., 2021).

## 2.3 UNCERTAINTY ESTIMATION

Uncertainty estimation in machine learning involves quantifying the quality of predictions with respect to the confidence or to the model parameters. There are various approaches for uncertainty estimation including Bayesian (Welling & Teh, 2011; Blundell et al., 2015; Gal & Ghahramani, 2016; Dusenberry et al., 2020) and non-Bayesian (Lakshminarayanan et al., 2017; Liu et al., 2020; Van Amersfoort et al., 2020) methods. This important step towards evaluating prediction reliability can be designed with (1) probabilistic models to cover full probability distributions over predictions (e.g., using Bayesian Neural Networks (Welling & Teh, 2011) (BNN)), (2) bootstrap methods to evaluate variability on controlled subsets of data or controlled subsets of the model weights (e.g., Monte-Carlo Dropout (Gal & Ghahramani, 2016)), or (3) ensemble methods to combine multiple model predictions (e.g., Deep Ensembles (Lakshminarayanan et al., 2017)). Tangentially, uncertainty calibration also gains attention to tune these techniques for capturing the prediction distributions as close to the sample distributions. Information theoretic approaches to use entropy and mutual information for estimating uncertainty by information gain (e.g., (Krishnan et al., 2020)) or calibration methods to align prediction probabilities to sample frequencies (e.g., (Krishnan & Tickoo, 2020; Kose et al., 2022)) are widely used for this purpose.

## 3 METHODOLOGY AND SETUP

### 3.1 DEEPFAKE DATASETS AND GENERATORS

Although there are several deepfake datasets in the literature, there exists only two multi-source datasets with known generators, namely FaceForensics++ (Rossler et al., 2019) (FF) and FakeAVCeleb (Khalid et al., 2021) (FAVC). FF contains 1000 real and 5000 deepfake videos, each 1000 created by FaceSwap (Fac, b), Face2Face (Thies et al., 2016), Deepfakes (Dee, c), Neural Textures (Thies et al., 2019), and FaceShifter (Li et al., 2019), presenting a representative dataset covering various aforementioned face manipulation methods. FAVC contains unbalanced number of real and fake videos created by FaceSwapGAN (Fac, a), FSGAN (Nirkin et al., 2019), and Wav-to-Lip (Prajwal et al., 2020). As real class has the lowest number of videos (500), we balance our setup by randomly selecting 500 videos from each class. We utilize FF as our main dataset and use FAVC for generalization, using 70/30 train/test splits for all detectors. Lastly, for the adversarial robustness experiment, we use a simple adversarial generator as outlined in Saremsky et al. (2022) on all subsets of FF where the black-box attack model is selected as the ResNet18 detector.

### 3.2 DEEPFAKE DETECTORS

In this paper, we run our uncertainty experiments with six deepfake detectors used across industry and academia. We select these as representatives from their family of detectors to keep the number of detectors tractable (i.e., Inception (Szegedy et al., 2016) is in the family of Xception (Chollet, 2017), ShuffleNet (Zhang et al., 2018) is in the family of MobileNet (Sandler et al., 2018), etc.).

- ResNet18 (He et al., 2016): a small and generic blind detector
- Xception (Chollet, 2017): most widely used generic blind detector with complex architecture (ffb)
- EfficientNet (Coccomini et al., 2022): one of the highest scoring deep blind detectors (ffb)
- MobileNet (Sandler et al., 2018): a compact blind detector with complex architecture
- FakeCatcher (Çiftçi et al., 2020a): an industry-adopted biological detector (nsa)
- Motion-based detector (Çiftçi & Demir, 2022): one of the newest biological detectors online.

The first four detectors consume raw data whereas the last two detectors exploit intermediate representations. FakeCatcher (Çiftçi et al., 2020a) extracts photoplethysmography (PPG) maps from

videos, representative of spatial, temporal, and spectral signal behavior of heart rates. We follow their construction of PPG maps, however we select segment duration as $\omega = 64$. Motion-based detector (Çiftçi & Demir, 2022) extracts dual-motion representations from videos to represent sub-muscular motion by deep and phase-based motion magnification. We follow their construction of motion tensors with the optimum suggested parameters. Intermediate representations for three types of detectors (raw, PPG-based, and motion-based) are sampled in App. 3. For network counterparts of the biological detectors, we use VGG19 (Simonyan & Zisserman, 2014) and C3D (Tran et al., 2015) respectively, as suggested in Çiftçi et al. (2020b) and Çiftçi & Demir (2022). However, our version of the motion-based detector uses clip duration of $\omega = 8$ frames instead of $\omega = 12$ frames due to differences in our C3D architecture implementation.

Deepfake detection studies only fake and real classes, where fake class equals to one source subset if it is a per-generator experiment, else covers samples of all generators. Source detection studies number of generators plus one classes (for real class) in total, which is formulated as classification.

### 3.3 UNCERTAINTY ESTIMATION

For our analysis, we employ Bayesian Neural Networks (Welling & Teh, 2011) to extend deterministic deep neural network architectures to corresponding Bayesian form in order to perform stochastic variational inference. This inference captures certainty measures that help us better understand the quality of predictions. Alternatively, we also apply MC Dropout (Gal & Ghahramani, 2016), which is another widely used Bayesian approximation for similar prediction analysis. Performance of both methods depend on multiple parameters, set optimally by our ablation studies.

BNN conversion of all models is achieved using Bayesian-torch repo (Krishnan et al., 2022). In order to help training convergence of models, we use MOPED method (Krishnan et al., 2020), which enables initializing variational parameters from a pretrained deterministic model. The model applies KL (Kullback-Leibler divergence) loss in addition to the cross entropy loss, scaling of which is controlled by $kl_{factor}$ parameter.

During inference, multiple stochastic forward passes are performed over the network via sampling from posterior distribution of the weights (with $n$ MC samples). Given a distribution of input features $x$ and labels $y$ over a dataset $D = \{x_j, y_j\}_{j=1}^{M}$, we first measure predictive uncertainty (predictive entropy, Eq. 1) capturing a combination of both data uncertainty and model uncertainty, which represents the uncertainty of the entire distribution. Then, we measure model uncertainty (mutual information between label $y$ and model parameters $\omega$, Eq.2) by computing the difference between the entropy of the expected distribution and the expected mean entropy of the ensembles.

$$H(y|x, D) := -\sum_{i=0}^{K-1} (p_{i\mu} \cdot log(p_{i\mu})) \tag{1}$$

$$I(y, \omega|x, D) := H(y|x, D) - E_{p(\omega|D)}[H(y|x, D)] \tag{2}$$

where $p_{i\mu}$ is predictive mean probability of $i^{th}$ class from $n$ MC samples, $\omega$ represents model parameters, and $K$ is number of output classes. Similar uncertainty decompositions are used by Depeweg et al. (2018) and Malinin & Gales (2018). In MC dropout experiments, we report model uncertainty as the mean of the variance of sampling outputs. Finally, model calibration analyses are conducted using retention plots for deepfake detection tasks.

### 3.4 PIXEL-WISE UNCERTAINTY

One of the most prominent techniques in Explainable AI has been saliency maps (Selvaraju et al., 2017), tracing the gradients back to input pixels to understand which pixels contribute more to the model's decision. However, saliency maps do not contain information about *how certain* this contribution is. We propose uncertainty maps to visualize this information to relate the model uncertainty back to generative artifacts on images. This duality can be thought analogous to having density plots in addition to retention plots for observing the model uncertainty with respect to its accuracy. We propose two types of maps: (1) saliency maps derived from Bayesian version of regular detectors, and (2) uncertainty maps tracing the uncertainty back to pixels of original images.

### 3.4.1 BAYESIAN SALIENCY MAPS

Saliency is computed in the traditional way by calculating a weighted average of penultimate layer activation maps, however using the BNN-converted versions of the aforementioned detectors.

$$\alpha_k = \frac{1}{n} \sum_n y_{\max} \left( \frac{1}{Z} \sum_{i,j} \frac{\partial y_{\max}}{\partial A^k_{ij}} \right), \; S = ReLU(\sum_k \alpha_k A^k) \tag{3}$$

The $\alpha_k$ activation weights are calculated as the pooled gradient magnitude of the $k^{th}$ activation map $A^k$, scaled by the predictive confidence $y_{\max}$ of the model, and averaged over the $n$ MC samples provided to the model, and computing the final saliency map $S$ by a linear combination of the $A^k$ activations with respect to $\alpha_k$ activation weights.

### 3.4.2 UNCERTAINTY MAPS

Although the previous approach pulls regular saliency maps towards uncertainty-informed saliency maps, they still do not represent pure uncertainty distribution on the input images. Thus, we formulate uncertainty maps by calculating predictive uncertainty over MC samples, and then map the gradient information from the predictive uncertainty back to input pixels. We define per-pixel uncertainty-based saliency in Eq. 4, following our notation in Eq. 1.

$$s_{ij} = \frac{\partial H(y|x, D)}{\partial x_{ij}} \tag{4}$$

## 4 RESULTS

We utilize the described setup to conduct experiments on uncertainty of deepfake (source) detectors, region- and pixel-based uncertainty, uncertainty estimation techniques, with ablation studies. Implementation details are documented in App. A.

### 4.1 UNCERTAINTY OF DEEPFAKE DETECTORS

Table 1: Accuracy and uncertainty results of regular/Bayesian detectors, evaluated per source on FF.

| Models | metrics | DF | F2F | FSh | FSw | NT | All |
|---|---|---|---|---|---|---|---|
| Resnet18 | accuracy (%) | 96.96 | 93.96 | 99.15 | 92.35 | 94.51 | 94.08 |
| BNN_Resnet18 | accuracy (%) | 96.38 | 94.91 | 98.81 | 93.64 | 93.39 | 95.32 |
| | predictive uncertainty | 0.075 | 0.077 | 0.043 | 0.097 | 0.069 | 0.037 |
| | model uncertainty | 0.031 | 0.028 | 0.026 | 0.051 | 0.038 | 0.018 |
| EfficientNet-B4 | accuracy (%) | 99.96 | 99.28 | 99.08 | 99.42 | 99.67 | 99.38 |
| BNN_EfficientNet-B4 | accuracy (%) | 95.87 | 93.24 | 98.82 | 97.75 | 92.31 | 90.93 |
| | predictive uncertainty | 0.209 | 0.151 | 0.203 | 0.168 | 0.246 | 0.263 |
| | model uncertainty | 0.098 | 0.092 | 0.107 | 0.095 | 0.132 | 0.128 |
| FakeCatcher | accuracy (%) | 96.73 | 95.12 | 95.65 | 96.04 | 93.31 | 96.14 |
| BNN_FakeCatcher | accuracy (%) | 96.30 | 94.37 | 95.52 | 95.76 | 91.59 | 95.77 |
| | predictive uncertainty | 0.015 | 0.026 | 0.056 | 0.016 | 0.089 | 0.028 |
| | model uncertainty | 0.001 | 0.003 | 0.008 | 0.002 | 0.006 | 0.002 |
| Motion-based Detector | accuracy (%) | 97.54 | 93.50 | 97.63 | 97.71 | 87.83 | 88.19 |
| BNN_Motion-based Detector | accuracy (%) | 94.16 | 84.91 | 95.91 | 92.95 | 77.71 | 87.40 |
| | predictive uncertainty | 0.083 | 0.147 | 0.071 | 0.103 | 0.241 | 0.182 |
| | model uncertainty | 0.007 | 0.007 | 0.008 | 0.007 | 0.006 | 0.007 |

Tabs. 1 and 2 show binary classification results for each generator in FF and FAVC. As an example, results in DF column are obtained by training and testing with DF and real class data with the aforementioned splits. Results in the last column of Tab. 1 and in Tab 2 are obtained using all fake classes and the real class for binary classification. We observe that, even if a complex network (EfficientNet) may produce high-accuracy results (99.38%), its BNN-version significantly drops in accuracy (90.93%), whereas simpler and biological detectors drop by less than 1% (rows 7 vs. 8 and

10 vs. 11). This phenomena is also reflected in predictive uncertainties (0.263 PU vs. 0.035 PU) meaning that complex detector is "more surprised" to see test samples. Model uncertainties show that there is more variation between ensembles of the complex network (0.0105), less variation for simpler network (0.035), and almost no variation for biological detectors (0.004 and 0.007), which sets a strong ground for biological detectors actually being able to capture the distribution well, as consistently observed in Tabs. 2, 3, 5.

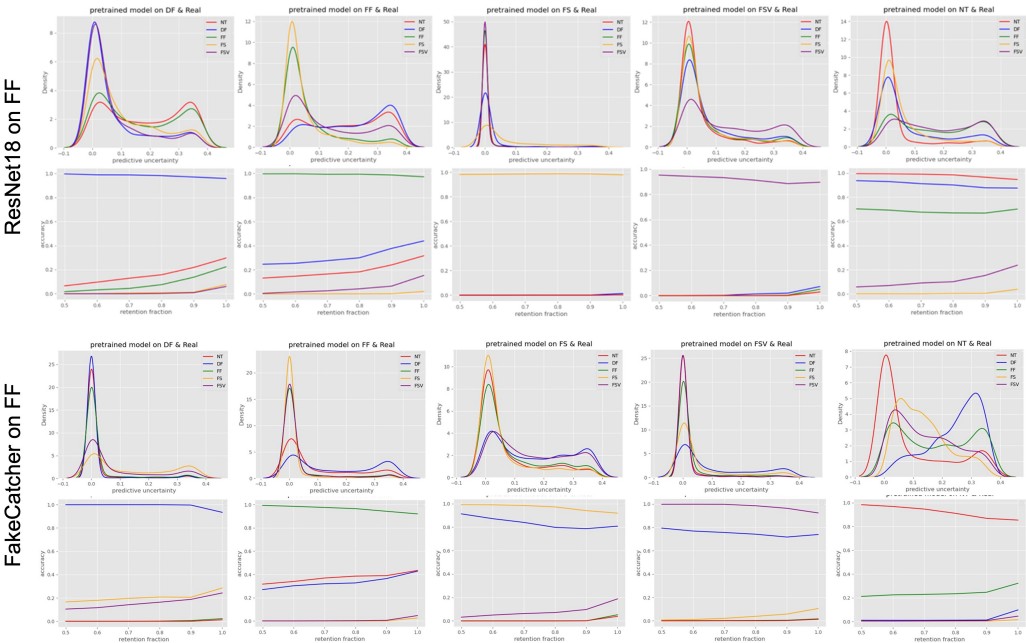

Figure 1: Each column shows density histograms and corresponding accuracy retention curves for five different generators in FF dataset trained and tested per generator, with Resnet and FakeCatcher.

Fig. 1 shows density histograms and retention curves, computed by testing ResNet18 and Fake-Catcher for real/fake detection with five generators in FF. Corresponding numeric results are reported in App. C. Retention fraction represents percentage of retained data based on predictive uncertainty, with the expectation of uncertain samples decreasing accuracy for calibrated models. We observe that (1) biological detectors have a narrower variance of uncertainty in this binary setting, (2) similar face manipulations provide relatively better generalizability for detectors (closer curves for DF, FSh, and FSw vs. NT and F2F) and (3) for biological detectors, per-generator trained models can generalize to similar generators' fakes (higher curves for FSw and FSh in columns 3 and 4 of the last row). For both detectors, we observe a unique behavior for NT, signalling that its generative residue has a significantly different distribution.

Table 2: Accuracy and uncertainty results of regular/Bayesian blind/biological detectors on FAVC.

| Resnet18 | accuracy (%) | 94.23 | FakeCatcher | accuracy (%) | 97.99 |
|---|---|---|---|---|---|
| BNN_Resnet18 | accuracy (%) | 93.54 | BNN_FakeCatcher | accuracy (%) | 98.21 |
| | predictive uncertainty | 0.119 | | predictive uncertainty | 0.030 |
| | model uncertainty | 0.054 | | model uncertainty | 0.009 |

We conduct leave-one-out (LOO) experiments (Tab. 3) for exploring generalizability further, with five training and one testing generator setups. Overall, generalizing to FSw's artifacts is harder, however FakeCatcher can achieve it. Another interpretation of Tab. 3 is that model uncertainty decreases (0.1116, 0.0094, 0.0092) and generalization capability increases (64.98%, 72.46%, 77.92%) as detectors use more modalities, from spatial (blind) to spatio-temporal (motion) to spectro-temporal (PPG) representations. Predictive uncertainty, on the other hand, does not seem to significantly vary

between models as all models are equally surprised to see out-of-domain samples. Retention plots for the first column is demonstrated in App.. F.

Table 3: Accuracy and uncertainty results of Bayesian versions of ResNet18, FakeCatcher, and motion-based detector with leave-one-out trainings per generator subset of FF.

| Models | metrics | $\text{LOO}_{DF}$ | $\text{LOO}_{F2F}$ | $\text{LOO}_{FSh}$ | $\text{LOO}_{FSw}$ | $\text{LOO}_{NT}$ |
|---|---|---|---|---|---|---|
| BNN_Resnet18 | accuracy (%) | 97.75 | 91.25 | 46.75 | 18.25 | 70.92 |
| | predictive uncertainty | 0.074 | 0.179 | 0.246 | 0.193 | 0.252 |
| | model uncertainty | 0.036 | 0.139 | 0.142 | 0.090 | 0.151 |
| BNN_Motion | accuracy (%) | 93.91 | 64.5 | 82.17 | 52 | 69.75 |
| | predictive uncertainty | 0.236 | 0.271 | 0.271 | 0.256 | 0.253 |
| | model uncertainty | 0.008 | 0.009 | 0.011 | 0.009 | 0.010 |
| BNN_FakeCatcher | accuracy (%) | 96.14 | 70.27 | 71.91 | 83.43 | 67.86 |
| | predictive uncertainty | 0.143 | 0.219 | 0.231 | 0.225 | 0.215 |
| | model uncertainty | 0.003 | 0.015 | 0.008 | 0.007 | 0.013 |

## 4.2 UNCERTAINTY MAPS FOR DEEPFAKE DETECTION

Following Sec. 3.4, we visually compare saliency map, Bayesian saliency, and uncertainty map of ResNet detector in Fig. 2, for NT (with the rest visualized in App. E). Saliency maps converge to obvious artifacts as expected, whereas Bayesian saliency tend to over-average important regions, creating *blob-like* areas in the middle. This is also expected as nose and mouth regions contain most of the artifacts. Uncertainty maps, however, create *skull-like* phantoms, because cheek, chin, and forehead areas contain less artifacts, increasing uncertainty. Comparing different generator artifacts, uncertainty maps help visualize that cheekbones are mostly smooth for NT, or chins are left unchanged by DF.

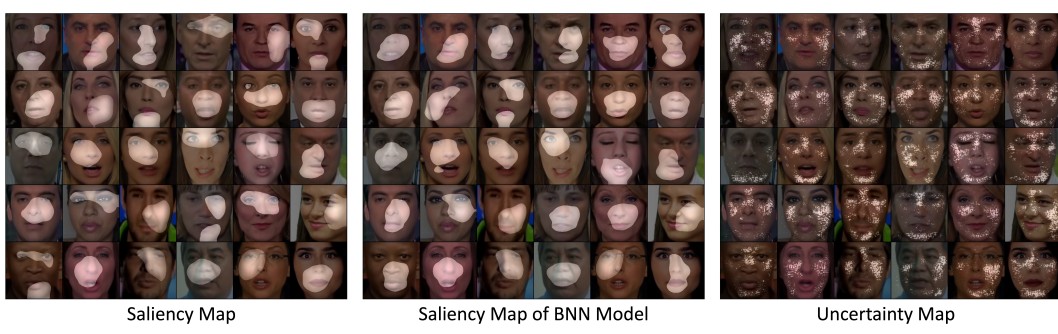

| Saliency Map | Saliency Map of BNN Model | Uncertainty Map |

Figure 2: Saliency, Bayesian saliency, and uncertainty maps of ResNet18 detector on NT samples.

## 4.3 UNCERTAINTY OF DEEPFAKE SOURCE DETECTION

Tab. 4 shows uncertainty and accuracy results for deepfake source detection on FF. Similarly, complex and large networks overfit and their Bayesian versions cannot reproduce the same accuracy (9.26% average decrease), with high model and predictive uncertainties (average 0.165 MU and 0.797 PU). Smaller networks (ResNet18, first block) and biological detectors (FakeCatcher, last block) preserve accuracies (0.73% average decrease) in the Bayesian setting and create more certain models (average 0.044 MU and 0.369 PU).

From a generator perspective, FSh Li et al. (2019) causes highest uncertainties, which can be explained by being the newest generative model with a complex distribution. We also observe that the per-model uncertainty distribution of specific detectors (e.g., MU and PU buckets for BNN Xception) is distinct enough to be utilized as an additional signal for source detection.

Table 4: Accuracy and uncertainty results of regular and Bayesian detectors for source detection on FF.

| Models | Metrics | DF | F2F | FSh | FSw | NT | All |
|--------|---------|-----|------|------|------|------|------|
| Resnet18 | accuracy (%) | 98.58 | 96.66 | 98.16 | 95.5 | 92.66 | 96.32 |
| BNN_Resnet18 | accuracy (%) | 97.83 | 97.41 | 97.33 | 96.75 | 93.66 | 95.95 |
| | predictive uncertainty | 0.055 | 0.068 | 0.134 | 0.103 | 0.108 | 0.131 |
| | model uncertainty | 0.024 | 0.033 | 0.082 | 0.060 | 0.055 | 0.074 |
| Xception | accuracy (%) | 99.83 | 99.41 | 98.92 | 99 | 99 | 99.17 |
| BNN_Xception | accuracy (%) | 97.08 | 98.25 | 91.41 | 95.58 | 99.91 | 89.37 |
| | predictive uncertainty | 0.257 | 0.109 | 0.518 | 0.388 | 0.054 | 0.344 |
| | model uncertainty | 0.160 | 0.075 | 0.366 | 0.266 | 0.031 | 0.227 |
| EfficientNet-B4 | accuracy (%) | 99.91 | 99.08 | 98.92 | 99.75 | 99.33 | 99.46 |
| BNN_EfficientNet-B4 | accuracy (%) | 82.75 | 88.75 | 85.75 | 94.33 | 92.5 | 89.77 |
| | predictive uncertainty | 0.984 | 0.806 | 1.091 | 0.714 | 0.844 | 0.894 |
| | model uncertainty | 0.437 | 0.444 | 0.571 | 0.382 | 0.372 | 0.432 |
| Mobilenetv2 | accuracy (%) | 99.91 | 99.33 | 99 | 99.58 | 99.08 | 99.38 |
| BNN_Mobilenetv2 | accuracy (%) | 87.41 | 91.83 | 93.58 | 97.91 | 84 | 91.08 |
| | predictive uncertainty | 1.328 | 1.021 | 1.210 | 0.824 | 1.259 | 1.154 |
| | model uncertainty | 0.326 | 0.449 | 0.535 | 0.402 | 0.447 | 0.447 |
| FakeCatcher | accuracy (%) | 92.08 | 90.31 | 92.41 | 90.97 | 85.27 | 91.26 |
| BNN_FakeCatcher | accuracy (%) | 93.58 | 88.34 | 93.20 | 91.67 | 80.98 | 90.18 |
| | predictive uncertainty | 0.125 | 0.223 | 0.122 | 0.139 | 0.310 | 0.198 |
| | model uncertainty | 0.004 | 0.015 | 0.006 | 0.005 | 0.032 | 0.013 |

## 4.4 Region-based Uncertainty Analysis

In order to couple face manipulation types to detector uncertainty, we conduct region-based experiments for source detection in Tab. 5. For example, removing symmetry elements from the training set (half mouth or one eye) reduces F2F source detection and increases uncertainty, as it is a mask-based technique creating symmetric priors. Logically, as information content decreases to half, uncertainties increase. Region-based results also indicate that uncertainty measures are highly correlated with accuracy measures, and lower face is more informative than upper face (higher accuracies and lower uncertainties). Deepfake detection version of this experiment is placed in App. D.

Table 5: Region-based analysis of uncertainty and accuracy for deepfake source detection on FF.

| Region | Models | metrics | DF | F2F | FSh | FSw | NT | All |
|--------|--------|---------|-----|------|------|------|------|------|
|  | Resnet18 | accuracy (%) | 96.25 | 95.25 | 97 | 90.83 | 88.83 | 93.65 |
| | BNN_Resnet18 | accuracy (%) | 95.25 | 93.33 | 98 | 91.33 | 92.75 | 93.25 |
| | | predictive uncertainty | 0.096 | 0.132 | 0.054 | 0.187 | 0.144 | 0.154 |
| | | model uncertainty | 0.050 | 0.062 | 0.029 | 0.092 | 0.067 | 0.076 |
|  | Resnet18 | accuracy (%) | 93.33 | 92.08 | 96.16 | 90.83 | 88.58 | 89.64 |
| | BNN_Resnet18 | accuracy (%) | 94.75 | 89.58 | 95.91 | 86.16 | 87.41 | 88.99 |
| | | predictive uncertainty | 0.140 | 0.192 | 0.112 | 0.267 | 0.192 | 0.217 |
| | | model uncertainty | 0.077 | 0.108 | 0.061 | 0.152 | 0.105 | 0.121 |
|  | Resnet18 | accuracy (%) | 95.42 | 91.5 | 97.58 | 93.33 | 88.16 | 91.88 |
| | BNN_Resnet18 | accuracy (%) | 94.33 | 90.75 | 97.75 | 91.16 | 87.58 | 91.37 |
| | | predictive uncertainty | 0.114 | 0.143 | 0.060 | 0.170 | 0.193 | 0.162 |
| | | model uncertainty | 0.057 | 0.073 | 0.031 | 0.085 | 0.103 | 0.083 |
|  | Resnet18 | accuracy (%) | 93.58 | 80.41 | 96 | 84.16 | 73.66 | 83.56 |
| | BNN_Resnet18 | accuracy (%) | 88.91 | 78.83 | 93.42 | 84.16 | 78.5 | 83.39 |
| | | predictive uncertainty | 0.168 | 0.277 | 0.167 | 0.254 | 0.266 | 0.247 |
| | | model uncertainty | 0.082 | 0.138 | 0.091 | 0.130 | 0.131 | 0.125 |

## 4.5 Ablation Studies

Tab. 6 shows the impact of parameters on BNN performance. In this table, Exp.1, Exp.2, Exp.3, Exp.4 and Exp.5 refer to the parameter settings of $n = \{40, 10, 40, 40, 40\}, \delta_{moped} =$

$\{0.1, 0.1, 0.5, 0.1, 0.1\}, kl_{factor} = \{1, 1, 1, 0.5, 0.1\}$ , respectively for number of MC samples $n$, moped delta value $\delta_{moped}$, and scaling coefficient for KL loss $kl_{factor}$. The results show that increasing $n$ from 10 to 40 does not have much impact on accuracy but improves the quality of uncertainty measures. Smaller $kl_{factor}$ causes degradation in performance in NT and the quality of uncertainty measures in general deepfake detection. Finally, increasing $\delta_{moped}$ causes a significant drop for BNN performance so it should be fine-tuned.

Tab. 7 shows that higher $dr$ (dropout ratio) causes significant drop in MC dropout performance as measured on two NT and F2F subsets. In contrast, a fine-tuned dropout ratio may improve accuracy.

Table 6: Impact of the selected hyperparameters on uncertainty performance.

| Dataset | Metrics | Exp. 1 | Exp. 2 | Exp. 3 | Exp. 4 | Exp. 5 |
|---|---|---|---|---|---|---|
| All | accuracy (%) | 96.72 | 96.94 | 87.79 | 96.60 | 97.09 |
| | predictive uncertainty | 0.042 | 0.052 | 0.227 | 0.058 | 0.050 |
| | model uncertainty | 0.025 | 0.026 | 0.120 | 0.036 | 0.029 |
| Neural | accuracy (%) | 93.61 | 93.79 | 84.50 | 93.91 | 90.92 |
| Textures | predictive uncertainty | 0.127 | 0.119 | 0.301 | 0.114 | 0.129 |
| | model uncertainty | 0.072 | 0.063 | 0.175 | 0.075 | 0.080 |

Table 7: Impact of the dropout ratio on resnet18 performance for binary classification.

| Metrics | NT | | | F2F | | |
|---|---|---|---|---|---|---|
| | dr=0.2 | dr=0.3 | dr=0.5 | dr=0.2 | dr=0.3 | dr=0.5 |
| accuracy (%) | 97.75 | 97.17 | 50.27 | 98.54 | 98.99 | 49.84 |
| model uncertainty | 0.015 | 0.026 | 0.030 | 0.013 | 0.011 | 0.023 |

## 4.6 Uncertainty on Adversarial Samples

Lastly, we measure the robustness of Bayesian detectors on adversarial samples. Tab. 8 reports accuracy of the attacked BNN ResNet18 before and after adversarial generation on five generator subsets. We select this model as it is the blind detector whose BNN version causes the lowest accuracy decrease. Based on the 93.53% average accuracy loss, we propose that more elaborate prevention mechanisms are needed against adversarial samples.

Table 8: Adversarial robustness of BNN Resnet detector.

| BNN_ResNet18 | DF | F2F | FSw | NT | FSh |
|---|---|---|---|---|---|
| Baseline accuracy | 95.81% | 95.69% | 89.58% | 93.64% | 97.26% |
| After adversarial attack | 0.30% | 0.03% | 3.07% | 0.94% | 0% |

## 5 Conclusion

We propose an in-depth analysis of deepfake detectors, generators, and source-detectors from an uncertainty perspective; including region-based detection experiments, novel uncertainty maps, blind and biological detector comparisons, and revelations between detector architectures and generator artifacts. Uncertainty analysis in the deepfake landscape is a new but essential dimension before releasing these detectors for public use. We have demonstrated that underconfident certain models are superior to overconfident uncertain models in terms of generalization. Our results indicate that generator artifacts can guide both detection and source detection, in image, region, and pixel levels.

As future work, we would like to build source detectors incorporating uncertainty maps directly into the classification process, as our experiments hinted such capacity.We also plan to expand our analysis to more comprehensive multi-source datasets such as ForgeryNet by He et al. (2021). As generative models and their applications become more ubiquitous and embedded in our everyday lives, we support the responsible dissemination of tools that foster explainability, transparency, trust, and risk-awareness to ensure their use for future social good.

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

## A    IMPLEMENTATION DETAILS

For our BNN implementations using Bayesian repo (Krishnan et al., 2022), prior parameters are set as $\mu_{prior} = 0$, $\sigma_{prior} = 1$, $\mu i_{posterior} = 0$, and $\rho i_{posterior} = -3$. *moped* is enabled with $\delta_{moped} = 0.1$ selecting *reparameterization* type.

During training, models are trained with Adam optimizer (Kingma, 2014) with a learning rate (LR) of 0.0001 for all architectures, except C3D. C3D LR is initiated as 0.001 and dynamic LR is applied with 0.1 scaling after each 10 epoch of overall 100 epochs. All other models are trained for 200 epochs. All weights are initiated using pretrained models on Imagenet (Russakovsky et al., 2015) from torchvision (Marcel & Rodriguez, 2010), except C3D model which is pretrained on UCF101 dataset (Soomro et al., 2012).

Model with the lowest validation loss is selected as the best model. Since accuracy is computed using $n$ MC samples, our definition of the best model may not always correspond to the model with the highest accuracy. MC sampling enables variations at the output that may cause some noise in accuracy. Predictive and model uncertainty results in the tables represent the average uncertainty measures of the test splits.

For saliency construction, batch size is selected as 1 and the saliency cut-offs are set as 20%, 20%, and 10% experimentally for saliency, Bayesian saliency, and uncertainty maps.

Finally, all reported accuracies are raw accuracies computed by running the final models on the corresponding intermediate data representations; thus, per-image and per-segment accuracies are not aggregated into per-video accuracies as performed by most of the detectors. Although this is a regular part of video-based detection, we did not incorporate this aggregation step in order to refrain from adding another layer of variables into our analyses.

## B    DATA REPRESENTATIONS

Fig. 3 include samples of intermediate data representations used by blind detectors, FakeCatcher, and motion-based detector; as raw face images, PPG maps, and motion tensors; for a real (top) and fake (bottom) video pair in FF from DF generator.

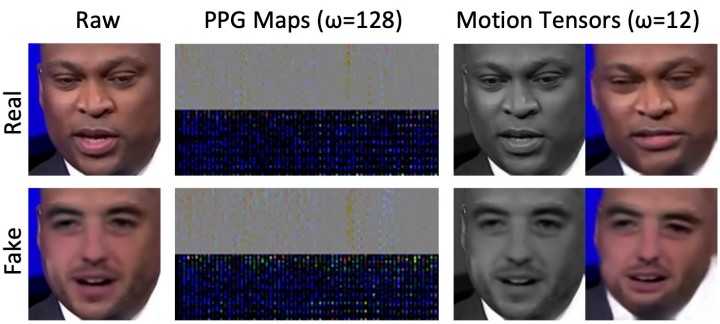

Figure 3: Data representations created by different deepfake detectors for a pair of real and fake videos from FF dataset's DF subset.

## C    PER-GENERATOR DEEPFAKE DETECTION

We document Bayesian ResNet18 and Bayesian FakeCatcher accuracies and uncertainties when the model is trained on one generator (and real class) and tested on another generator in Tabs. 4. While the results on the diagonal are high as expected, cross-generator generalization highly varies between manipulation types, supporting our claims for Fig. 1. Nevertheless, biological detectors seem to support slightly better single-generator generalization (with higher accuracies and lower uncertainties).

**ResNet18** — Trained on Real and Generator Subset / Tested on Generator Subset

| | metrics | DF | F2F | FSh | FSw | NT |
|---|---|---|---|---|---|---|
| DF | accuracy (%) | 95.92 | 22.41 | 7.66 | 6.08 | 29.75 |
| | predictive uncertainty | 0.077 | 0.165 | 0.107 | 0.078 | 0.183 |
| | model uncertainty | 0.034 | 0.070 | 0.044 | 0.029 | 0.077 |
| F2F | accuracy (%) | 44.08 | 97.25 | 2.17 | 15.42 | 31.75 |
| | predictive uncertainty | 0.217 | 0.069 | 0.053 | 0.137 | 0.195 |
| | model uncertainty | 0.075 | 0.029 | 0.014 | 0.048 | 0.066 |
| FSh | accuracy (%) | 1.41 | 0.3 | 97.91 | 0.25 | 0.25 |
| | predictive uncertainty | 0.021 | 0.006 | 0.076 | 0.006 | 0.008 |
| | model uncertainty | 0.006 | 0.002 | 0.049 | 0.002 | 0.003 |
| FSw | accuracy (%) | 7.25 | 5.08 | 3.08 | 89.75 | 2.91 |
| | predictive uncertainty | 0.081 | 0.066 | 0.061 | 0.142 | 0.052 |
| | model uncertainty | 0.034 | 0.029 | 0.026 | 0.080 | 0.022 |
| NT | accuracy (%) | 87.58 | 70.25 | 3.91 | 23.83 | 94.66 |
| | predictive uncertainty | 0.090 | 0.169 | 0.067 | 0.178 | 0.040 |
| | model uncertainty | 0.048 | 0.095 | 0.036 | 0.103 | 0.022 |

**FakeCatcher** — Trained on Real and Generator Subset / Tested on Generator Subset

| | metrics | DF | F2F | FSh | FSw | NT |
|---|---|---|---|---|---|---|
| DF | accuracy (%) | 93.44 | 2.29 | 28.60 | 24.34 | 1.29 |
| | predictive uncertainty | 0.015 | 0.027 | 0.142 | 0.086 | 0.018 |
| | model uncertainty | 0.001 | 0.002 | 0.018 | 0.020 | 0.001 |
| F2F | accuracy (%) | 42.90 | 92.10 | 2.65 | 4.75 | 43.52 |
| | predictive uncertainty | 0.163 | 0.035 | 0.021 | 0.031 | 0.099 |
| | model uncertainty | 0.017 | 0.005 | 0.001 | 0.002 | 0.022 |
| FSh | accuracy (%) | 80.84 | 5.28 | 91.94 | 18.82 | 3.95 |
| | predictive uncertainty | 0.163 | 0.091 | 0.067 | 0.157 | 0.075 |
| | model uncertainty | 0.029 | 0.002 | 0.015 | 0.031 | 0.002 |
| FSw | accuracy (%) | 73.96 | 1.74 | 10.57 | 92.43 | 1.35 |
| | predictive uncertainty | 0.111 | 0.029 | 0.064 | 0.017 | 0.018 |
| | model uncertainty | 0.012 | 0.002 | 0.008 | 0.003 | 0.002 |
| NT | accuracy (%) | 9.79 | 32.23 | 1.54 | 4.57 | 85.45 |
| | predictive uncertainty | 0.243 | 0.180 | 0.136 | 0.146 | 0.096 |
| | model uncertainty | 0.003 | 0.017 | 0.001 | 0.002 | 0.011 |

Figure 4: Training ResNet18 and FakeCatcher on one generator subset and testing on another one.

## D REGION-BASED DEEPFAKE DETECTION

Similar to our region-based *source* detection results demonstrated in Tab. 4, Tab. 9 shows region-based *deepfake* detection results, in terms of uncertainty and accuracy. As the training sets now include single generators, generative artifacts dominate the impact of regions.

Table 9: Region-based analysis of uncertainty and accuracy for deepfake detection on FF.

| Region | Models | metrics | DF | F2F | FSh | FSw | NT | All |
|---|---|---|---|---|---|---|---|---|
| | Resnet18 | accuracy (%) | 99.45 | 98.20 | 99.30 | 99.10 | 97.76 | 97.86 |
| | BNN_Resnet18 | accuracy (%) | 98.37 | 97.69 | 99.17 | 97.03 | 92.17 | 95.00 |
| | | predictive uncertainty | 0.133 | 0.077 | 0.047 | 0.103 | 0.124 | 0.058 |
| | | model uncertainty | 0.056 | 0.045 | 0.026 | 0.058 | 0.079 | 0.035 |
| | Resnet18 | accuracy (%) | 99.30 | 96.46 | 97.86 | 95.02 | 95.77 | 97.56 |
| | BNN_Resnet18 | accuracy (%) | 97.63 | 95.27 | 97.99 | 94.37 | 88.03 | 94.45 |
| | | predictive uncertainty | 0.085 | 0.099 | 0.053 | 0.101 | 0.145 | 0.080 |
| | | model uncertainty | 0.051 | 0.053 | 0.034 | 0.069 | 0.095 | 0.049 |
| | Resnet18 | accuracy (%) | 99.54 | 97.27 | 98.84 | 98.05 | 95.81 | 97.32 |
| | BNN_Resnet18 | accuracy (%) | 96.04 | 96.65 | 98.21 | 94.67 | 84.47 | 92.48 |
| | | predictive uncertainty | 0.101 | 0.111 | 0.087 | 0.115 | 0.154 | 0.089 |
| | | model uncertainty | 0.063 | 0.072 | 0.063 | 0.069 | 0.093 | 0.063 |
| | Resnet18 | accuracy (%) | 98.95 | 95.53 | 98.86 | 95.88 | 91.68 | 95.08 |
| | BNN_Resnet18 | accuracy (%) | 96.64 | 91.68 | 98.19 | 92.58 | 79.80 | 83.78 |
| | | predictive uncertainty | 0.093 | 0.127 | 0.044 | 0.106 | 0.140 | 0.250 |
| | | model uncertainty | 0.055 | 0.076 | 0.030 | .0.067 | 0.088 | 0.054 |

## E PIXEL MAPS OF ALL GENERATORS

Saliency, Bayesian saliency, and uncertainty maps demonstrated for NT in Fig. 2 is expanded in Fig. 5 to cover all generator subsets of FF. As mentioned, different generative artifacts attended by the detector are best visualized by uncertainty maps (last column), guiding our future work.

## F RETENTION PLOTS FOR LOO EXPERIMENTS

Fig. 6 visualizes retention curves for the first column of Tab. 3. Here, $LOO_{DF}$ (blue curve) refers to the use of F2F, FSh, FSw, NT, and real data in training and validation, and DF data only for

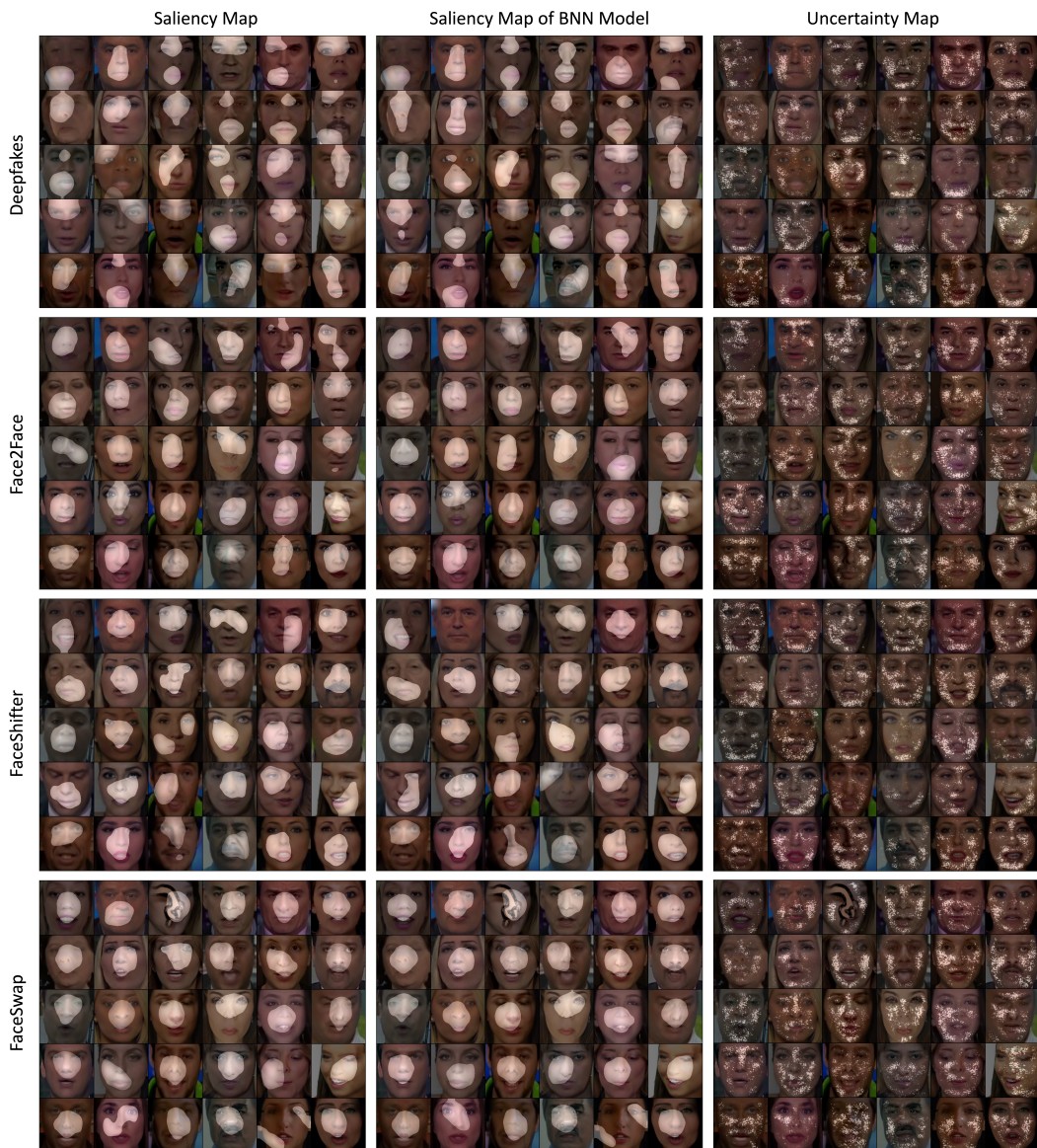

Figure 5: Saliency, Bayesian saliency, and uncertainty maps of ResNet18 detector on all generator subsets of FF.

testing. Each column shows retention curves for the corresponding model in Tab. 3. Retention curves confirm our prior findings about generalization.

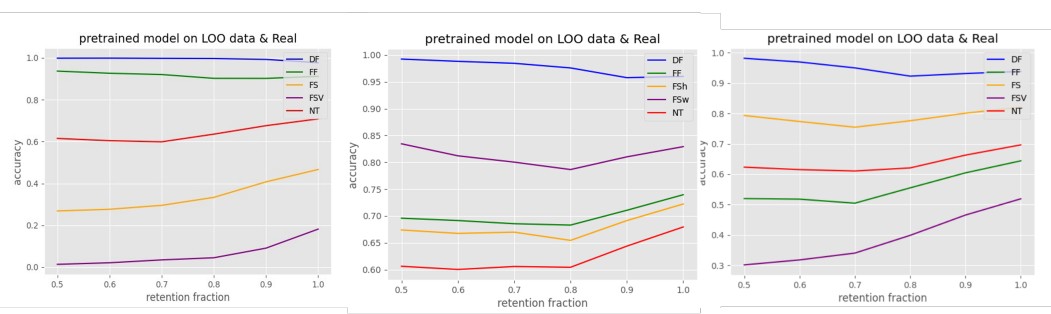

Figure 6: Retention plots of (a) ResNet18, (b) FakeCatcher, (c) motion-based detector on FF for DF.

