# OpenReview forum: "ARE YOU CERTAIN THAT IT IS A DEEPFAKE? DETECTION, GENERATION, AND SOURCE DETECTION FROM AN UNCERTAINTY PERSPECTIVE"
_ICLR.cc/2024/Conference — Submitted to ICLR 2024_

### Official Review · Reviewer_eCwJ · 2023-10-20

**Soundness:** 3 good
**Presentation:** 2 fair
**Contribution:** 2 fair
**Rating:** 3
**Confidence:** 4

**Summary:**

This paper analyzes the uncertainty of various AI models for deepfake detection. In particular, the authors propose two uncertainty metrics based on established Bayesian frameworks (Bayesian neural networks and Monte Carlo dropout) to quantify the confidence of the deepfake detectors' judgments. These metrics are used to compare the in-domain and cross-domain generalization ability of different model architectures, i.e., convolutional networks and detectors of biological signals. The authors further conduct leave-out-one experiments, extract uncertainty-driven saliency maps for explainability, and analyze the robustness against adversarial attacks. The paper’s main conclusion is that underconfident certain models are superior to overconfident uncertain models in terms of generalization.

**Strengths:**

The main strength of the paper is the problem formulation and the angle of study, i.e. to focus on the uncertainty of deepfake detection models, which is a great concern when deploying in real-world systems. Estimating the uncertainty of deepfake detectors is indeed a worthwhile goal that is underexplored in the literature. The Bayesian techniques for uncertainty quantification that are used in the paper are emerging in the literature and have sufficient theoretical justification. They truly have the potential to restore trust in AI models and enhance the reliability of deepfake forensics.

**Weaknesses:**

Despite the promising idea, the authors do not convincingly and clearly show the value of the proposed metrics and the goal of the uncertainty analysis. First, the meaning of the metrics is not explained and must be found in the literature, e.g., in the similar setting of Ranganath et al, which is not cited near the definitions of the metrics. I understand that the predictive uncertainty captures the concentration of the predictive distribution after we average the distributions of the sampled models, while the mutual information captures the consistency of the distributions of the sampled models. It is difficult for a user to draw meaningful conclusions about a model’s output from such metrics, at least not without a straightforward interpretation and calibration  This analysis is more appropriate for model comparison but does not clarify which metric is better suited to the task, while it can be argued that the paper conclusions are not supported by the results

Recall that the paper’s main conclusion is that underconfident certain models are superior to overconfident uncertain models in terms of generalization. Table 1 evaluates various models and their Bayesian counterparts on the in-domain generalization task, i.e., the models are trained on FaceForensics and are evaluated on the unseen test set of  the same dataset . The authors conclude that, since the accuracy of the blind detectors drops more when switching to the Bayesian setting (which incorporates model uncertainty) compared with the biological detectors, the blind detectors are more prone to overfitting and this is reflected in the uncertainty metrics. From the two blind models of the experiment however only the accuracy of the BNN_EfficientNetB4  drops, while the accuracy of the BNN_Resnet18 increases. Even for BNN_EfficientNetB4, the conclusion is hard to justify. For example, the 99.38% accuracy of the deterministic EfficientNetB4 drops to the 90.93% of the BNN_EfficientNetB4 on the unseen test set of FF. Does this mean that the deterministic model overfits or that it generalizes better than the Bayesian model in the same domain?  The BNN_EfficientNetB4 with accuracy 90.93% also has higher uncertainty than the BNN_Motion detector with accuracy 87.40%. Does this mean that the higher test accuracy of the BNN_EfficientNetB4 is less reliable than the other model’s?

If overfitting here is meant in the context of cross-domain generalization, no conclusions can be drawn because no cross-domain results are reported for EfficientNet B4. On the other hand, both BNN_Resnet18 and BNN_FakeCatcher generalize with >90% accuracy to FAVC despite BNN_Resnet18 having higher uncertainty than BNN_FakeCatcher in Table 1.The authors claim that BNN_Resnet18 has higher uncertainty on FAVC compared with BNN_FakeCatcher but does this imply that the 93.54% accuracy of BNN_Resnet18 on the unseen domain is less reliable? These concerns are even more striking in the leave-one-out experiments of Table 3; when BNN_Resnet18 and BNN_Motion are evaluated on the unseen FSw manipulation, their accuracies drop considerably to 18.25% and 52% respectively, yet no appreciable difference is reflected in the uncertainty metrics. Similar observations can be made for the cross-domain results of Appendix C.

A further weakness of the paper is its poor writing, which is without spelling mistakes but  frequently confusing. For example, different groups of models with different training setups are compared in different experiments. Crucial details are omitted or  difficult to find, e.g., the fact that detection is studied at the frame level and aggregation is not  performed is stated in Appendix A, retention curves are not explained etc. Plots are also confusing, e.g., Figure 1 is difficult to read and the values of mutual information seem to extend in negative values, adjacent plots within the same figure do not have the same y-axis, etc. GANs are mentioned extensively in the introduction but concern only 2 out of the 5 manipulations of FaceForensics.

As final comments, while the authors state that they chose a representative collection of detectors, all blind detectors are convolutional-based and all detector models operate at the frame level. Newer architectures tend to operate at the video level directly and use more complex architectures such as 3D-convolutional, LSTM, and vision transformers. The authors also claim that FaceForensics and FAVC are the only datasets that reveal the manipulation method; however, ForgeryNet is another choice that is substantially larger and more diverse.

Based on the poor interpretation of the results and writing of the paper, my recommendation is unfortunately to reject the paper, despite its very promising focus. I would like to encourage the authors to rethink the usage of the uncertainty metrics, refocus their objectives, and resubmit their improved work.

**Questions:**

See my detailed comments in the weaknesses section.

---

### Official Review · Reviewer_DTE3 · 2023-10-31

**Soundness:** 2 fair
**Presentation:** 2 fair
**Contribution:** 1 poor
**Rating:** 3
**Confidence:** 4

**Summary:**

The authors introduce various uncertainty metrics to evaluate the performance of deepfake detectors. Specifically, they employ a Bayesian approach for numerical uncertainty evaluation and present two visualization methods for uncertainty assessment. These methods are subsequently applied to two renowned deepfake datasets for validation.

**Strengths:**

•	The paper is articulately written, making it easy for readers to understand the presented content.

•	The authors provide numerous visualizations throughout the paper, aiding in comprehension and the conveyance of key points.

•	The study covers multiple uncertainty metrics to deliver a comprehensive evaluation.

**Weaknesses:**

•	The paper's contributions appear somewhat limited. While the authors present their evaluations and experiences, there seems to be a lack of deeper analytical insights.

•	The sole reliance on FAVC for generalization evaluation appears to be an uncommon choice, especially when there are other prominent datasets available like CelebDfv2, DFDC, and WildDeepfake.

•	Considering the scope of this study, the number of deepfake generation methods and detectors analyzed seems limited.

•	Previous work has introduced certain metrics that the authors seem to reuse. Given the limited datasets and models used in this study, the novelty and depth of insights appear constrained.

•	It would be beneficial for the authors to emphasize key findings in each experiment to guide readers through the paper's narrative more effectively.

**Questions:**

•	Could the authors clarify the term "blind detector" as mentioned in the manuscript? An explanation or definition would help in understanding its context.

•	I would appreciate it if the authors could address the concerns and weaknesses outlined above.

---

### Official Review · Reviewer_Aw4C · 2023-11-01

**Soundness:** 2 fair
**Presentation:** 1 poor
**Contribution:** 2 fair
**Rating:** 3
**Confidence:** 4

**Summary:**

This paper applies Bayesian Neural Networks to measure the uncertainty of deepfake detectors. A pixel level uncertainty map can also be obtained by the gradient of predictive uncertainty over pixels. Uncertainty for multiple deepfake detectors are computed under different experiment setting.

**Strengths:**

Uncertainty of deepfake detectors is a relatively less investigated area.

**Weaknesses:**

1) The motivation of this paper is not well justified. In the introduction, the authors mentioned "Understanding model response
... help compare their generalization in-the-wild, their overfitting to the artifacts, their performance beyond the training distribution, their robustness against adversarial attacks, and their effectiveness in source detection." However, the paper does not provide enough information about any of them. Section 4.3 and 4.6 talk about source detection and adversarial samples, but both sections are too short and not informative, so it is unclear how the predictive uncertainty help these two tasks?

2) The technical contribution is quite low, since the major technique is to adopt BNN (a technique from 12 years ago).

3) How do we know whether predicted uncertainty of deepfake detectors is accurate? In other words, how to evaluate uncertainty prediction in the case of deepfake detectors?

4) The paper is poorly written. The technical part is not clearly presented: for example, in (1) and (2), most notations are not explained, e.g., what is x, y, D? There are lots of typos, grammar errors, and even broken sentences. For example, in page 4 above equation (1), "We measure predictive uncertainty (predictive entropy) capturing a combination of input uncertainty and model uncertainty, and model uncertainty (mutual information) computing the difference between the entropy of the mean of the predictive distribution and the mean of the entropy".  It is too hard to understand what the sentence is trying to say.

**Questions:**

See the weakness section.

With all three reviewers giving low ratings, the authors came up with a rebuttal that did not provide sufficient meaningful information, but lots of aggressive languages. Not really professional.

---

### Author Response · Authors · 2023-11-23
**Rebuttal**

We thank reviewers for their detailed and constructive feedback, especially for finding our problem formulation novel and underexplored ($\color{red}{R1}$, $\color{blue}{R3}$), our paper articulately written ($\color{green}{R2}$), our visualizations helpful ($\color{green}{R2}$), used Bayesian techniques and multiple uncertainty metrics justified and comprehensive ($\color{green}{R2}$, $\color{blue}{R3}$), and for mentioning our approach’s potential to restore trust in AI ($\color{blue}{R3}$). Below, we respond with clarifications and corresponding changes in our revision. Trivial fixes and additions are not listed but changed in the revised version.

**Contributions ($\color{red}{R1}$, $\color{green}{R2}$):** We placed some observations and outcomes back to Sec. 4.1, 4.3 and 4.4. which were initially cut for space constraints. For each result, we emphasized and quantifiably supported the key finding. We added mentions of our novel uncertainty maps and explainability comparisons in the introduction. We also refined our research questions.

**Clarification on low contribution due to uncertainty methods ($\color{red}{R1}$):** We do not only “adopt BNNs from 12 years ago ($\color{red}{R1}$)”. As mentioned by other reviewers, we deliver “a comprehensive evaluation using multiple uncertainty metrics ($\color{green}{R2}$)”, which are “emerging in the literature and have sufficient theoretical justification ($\color{blue}{R3}$)”. To enumerate, we utilize model and predictive uncertainties, based on BNNs and Monte-Carlo dropout, compared and evaluated by density and retention plots, supported by saliency and novel uncertainty map visualizations.

**Accuracy of predictive uncertainty ($\color{red}{R1}$):** Model calibration analysis (goodness of predictive uncertainty) is done through density and retention plots in Fig. 1, also supported by the ablation studies in Sec. 4.5.

**Clarity on Eqns. 1 and 2 ($\color{red}{R1}$, $\color{blue}{R3}$):** We thank $\color{red}{R1}$ for fixing our assumption about the reader having basic knowledge of probability distributions. We defined x, y, D, and $\omega$. We simplified the mentioned sentence, added what those metrics mean and references formulating uncertainty decompositions similarly.


**Clarification about datasets ($\color{green}{R2}$, $\color{blue}{R3}$):** We focus on known multi-source datasets in order to analyze the generator-detector ecosystem (see LOO and per-generator experiments). CelebDF is single-source, WildDeepfake is unknown source, and DFDC labels are problematic. We added ForgeryNet as future work.

**Number of detectors ($\color{green}{R2}$, $\color{blue}{R3}$):** Our full set of experiments indeed cover other detectors. However, current 6 detectors with reasonings (Sec. 3.2) are selected to represent their family of detectors, so that we can induce and summarize our findings fitting to 9 pages. We can add SqueezeNet, Inceptionv3, MesoNet, DenseNet, and others to the list and the observations would not change – reducing all possible detectors to a few categorical detectors can indeed be considered as a side contribution of our paper. We agree with $\color{blue}{R3}$ that temporal detectors should also be explored such as 3D CNNs and CNN-LSTMs.

**Tab. 1 clarification ($\color{blue}{R3}$):** We note (and clarify in the paper) that ResNet here is not a complex network, as per Sec. 3.2. Predictive uncertainty numbers documented in text are the average of simpler and biological networks.

**Predictive/model uncertainty observations ($\color{blue}{R3}$):** We added all detailed observations back to the paper. In almost all detection tasks, biological detectors (followed by simpler networks) have much less model uncertainty than deep and complex networks, which supports our main claim.

**Clarification on generalization ($\color{blue}{R3}$):** As explained in Sec 4.1 and Appendix C, generalization is assessed by (i) overall drop in accuracy by the BNN version, (ii) closer curves in retention plots, (iii) training on one generator subset and testing on others, and (iii) leave-one-out experiments. Out of the four, last three can be considered cross-domain as generative residue and manipulation methods greatly vary between these subsets.

---

### Meta-Review · Area_Chair_xVSL · 2023-12-06

**Metareview:**

3x R. This paper proposes two uncertainty metrics based on Bayesian to quantify the confidence of deepfake detection. The reviewers converge to the common concerns about the (1) unclear writing, (2) shaky motivation, (3) insufficient technical contributions, and (4) limited testing data and target models, which are not addressed by the rebuttal. The AC therefore leans not to accept this submission.

**Justification For Why Not Higher Score:**

N/A

**Justification For Why Not Lower Score:**

N/A

---

### Decision · Program_Chairs · 2024-01-16

Reject